# From Data to Diagnosis: How Machine Learning Is Changing Heart Health Monitoring

**DOI:** 10.3390/ijerph20054605

**Published:** 2023-03-05

**Authors:** Katarzyna Staszak, Bartosz Tylkowski, Maciej Staszak

**Affiliations:** 1Institute of Chemical Technology and Engineering, Faculty of Chemical Technology, Poznan University of Technology, ul. Berdychowo 4, 60-965 Poznan, Poland; 2Eurecat, Centre Tecnològic de Catalunya, C/Marcellí Domingo s/n, 43007 Tarragona, Spain

**Keywords:** medicine, heart, machine learning, ECG, PPG

## Abstract

The rapid advances in science and technology in the field of artificial neural networks have led to noticeable interest in the application of this technology in medicine. Given the need to develop medical sensors that monitor vital signs to meet both people’s needs in real life and in clinical research, the use of computer-based techniques should be considered. This paper describes the latest progress in heart rate sensors empowered by machine learning methods. The paper is based on a review of the literature and patents from recent years, and is reported according to the PRISMA 2020 statement. The most important challenges and prospects in this field are presented. Key applications of machine learning are discussed in medical sensors used for medical diagnostics in the area of data collection, processing, and interpretation of results. Although current solutions are not yet able to operate independently, especially in the diagnostic context, it is likely that medical sensors will be further developed using advanced artificial intelligence methods.

## 1. Introduction

The year 2022 can be proudly called the year of the Holter monitor. Exactly 60 years ago, on 6 July 1962, Norman J Holter and Wilford R Glasscock filed a patent application for a heart rate (HR) monitor [1], and began its production in collaboration with Bruce Del Mara. However, this would not have been possible without Holter’s earlier work in 1947, when he made the first transmission of a radioelectrocardiogram (RECG). This required the use of an 85-pound piece of equipment that Holter strapped to his back while riding a stationary bicycle [2]. Since then, Holter monitors have evolved both in terms of the precision of measurements taken and a decrease in the size of the equipment. In 1982, Holter stated that “[t]he 85 lb. RECG, although not practical, represented a major breakthrough since before that time a patient had to lie quietly. Our greatest contribution was a radical one and was the beginning of an era where one could take ECG’s on skiers, parachute jumpers, runners, and just about any other type of vigorous physical activity” [3]. Today, medical heart sensors include not only ECGs but also photoplethysmography (PPG), which measures volumetric variations in blood circulation, and thereby, the heart rate. Regardless of the method of measurement, cardiac monitoring sensors are constantly being improved. Such improvements include more precise measurements, which can be taken in real time, and more rapid analysis of results. Many of these improvements have been achieved through the use of artificial intelligence (AI).

AI has become a permanent fixture, both in everyday life and in the scientific world. The field of AI research was established as far back as 1956 in the USA. Nowadays, this technology has become increasingly popular and, more importantly, it has become more precise thanks to the increased computing power of computers. This progress is expected to accelerate. AI plays a significant role in medicine, especially in the analysis of medical images [4], drug design [5], and prediction [6]. The key reasons for using AI in medicine are increased accuracy and improved efficiency, due to the fact that AI algorithms can process large amounts of medical data quickly. It should be noted that attempts have been made to use artificial neural networks (ANNs) in clinical evaluation [7,8] and in real-life implementation [9]. As ANNs develop, scientists are exploring new applications of this technology, for example, to save or monitor human lives. Therefore, a natural progression of this research was to apply this technology to medical sensors, mainly to improve the analysis of transmitted signals. Medical sensors and, by extension, diagnostics play a crucial role in accurately evaluating human health––and not only in patients undergoing treatment. Healthy people also want to monitor their vital signs; for example, recreational runners monitor their pulse rate during exercise [10]. Measurement of the HR in asymptomatic people can serve several purposes, including screening for underlying medical conditions, monitoring health status, assessing fitness levels, and evaluating the effectiveness of medications. HR measurements can be particularly important for people with risk factors for heart disease or other medical conditions. However, the decision to measure HR in asymptomatic people should be made on a case-by-case basis, taking into account individual risk factors and medical history. One example can be found in study [11], in which the authors investigated the relationship between health-related quality of life in the physical domain (HRQOLphysical) and heart rate variability (HRV) in healthy adults. The autonomic nervous system (ANS) was considered a potential mechanism that could link the two aforementioned considerations. The study found that low HRQOLphysical was independently associated with a reduction in HRV, which can increase the risk of cardiovascular disease. These findings suggest that the ANS may play a role in the relationship between HRQOLphysical and CVD. In addition, the monitoring of people without cardiac disease has proved useful in combatting COVID-19 [12]. Consumer products, such as wearable devices or mobile applications, can potentially identify early signs of illness by measuring health metrics, such as respiration rate, heart rate, or heart rate variability. These devices may prove a valuable tool in the fight against COVID-19; data from 2745 subjects diagnosed with COVID-19 showed that measuring physiological signs can help predict illness on a specific day, while self-reported symptoms can predict the need for hospitalization. Respiration rate and heart rate are generally elevated by illness, while HRV is decreased. Measuring these metrics, along with molecular-based diagnostics, may lead to better early detection and monitoring of COVID-19.

Precise measurements can ensure rapid diagnosis and clinical decision support, thereby not only saving lives but also reducing treatment costs [13]. According to the literature not only on medical sensors but also on signal analysis in general, systems based on machine learning have a high potential for improving analysis accuracy. Medical sensors can effectively assess human health in several ways. One such way is real-time monitoring, whereby sensors continuously monitor vital signs, providing real-time data on important health parameters, for example, heart rate, blood pressure, oxygen levels, and more. Further, many medical sensors are noninvasive, meaning that they do not need to penetrate the skin, thereby minimizing the risk of infection and discomfort for the patient. Additionally, the collection of objective data on the patient’s health can provide information to health care professionals to make informed decisions without relying solely on subjective information from the patient. The use of sensors also allows the early detection of signs of disease, achieved by continuously monitoring a patient’s health. Of course, medical sensors are just one of the tools available for assessing human health, thereby understanding a patient’s health status. Thus, using wearable sensors to monitor health can provide benefits to the general population and healthy individuals. Such benefits including, as mentioned above, the early detection of health problems before they become more serious. Wearable sensors provide feedback on physical activity levels, sleep patterns, and other lifestyle habits, allowing people to make changes to improve their health. Additionally, by allowing individuals to monitor their own health, wearable sensors can increase patient engagement in their own health and well-being. It should be noted that proper real-time diagnosis is an important aspect, especially today. The challenges faced by doctors during the COVID-19 pandemic––including the need to monitor the condition of patients during the course of the disease, both in hospital and during home quarantine––highlight the need for the development of medical sensors that are wearable, accurate, and allow real-time diagnosis [14,15].

The aim of this review was to evaluate the state-of-the-art application of machine learning methods to heart sensors. The review also aimed to determine priorities for future research in the rapidly progressing field of AI technology by identifying current trends and limitations in the development of medical sensors.

## 2. Methods

Bibliometric analysis was conducted using the PubMed, Scopus, and Google Scholar search engines, as well as the Espacenet and PatFT patent databases, supported by the European Patent Office and the United States Patent Trademark Office, respectively. The following combinations of keywords were used: “deep learning”, “machine learning”, “ML”, “artificial intelligence”, “AI”, “heart sensor”, “electrocardiography”, “ECG”, photoplethysmography”, and “PPG”. Articles were excluded if they were written in a language other than English, lacked ECG or PPG data, lacked machine learning methods, were published before 2018, and were presented as a cluster analysis of semantic web using VOSviewer software in the Appendix A (Appendix A). Exceptions were articles that were essential to the development of the techniques described, notably the early Holter work. At the same time, to ensure originality and accuracy, the search focused on research papers, while excluding review articles. As a result, papers with titles containing words such as “review”, “study”, “meta-analysis”, or “overview” were excluded.

The systematic literature review was conducted as follows:A.Planning by initial idea formulation: Medical sensors supported by ML.PICOC: Population: medical sensors, machine learning. Intervention: use of medical sensors supported by machine learning (ML) algorithms for monitoring patients’ health status. Comparison: use of medical sensors supported by ML algorithms compared with traditional approaches. Outcomes: accuracy and reliability of medical sensors supported by ML, clinical efficacy, patient outcomes, and cost-effectiveness. Context: challenges and opportunities associated with integrating sensors and ML algorithms into clinical practice.Research questions: i. How have medical sensors been used to improve patient outcomes? ii. How accurate and reliable are medical sensors supported by machine learning, and which factors influence their performance? iii. What are the most commonly used machine learning algorithms to analyze data collected from medical sensors? iv. How do medical sensors supported by machine learning impact health care delivery and outcomes, including patient satisfaction, clinical value, and cost-effectiveness? v. What are the legal and regulatory implications of using medical sensors supported by machine learning?Digital library sources: Google Scholar, Scopus, PubMed, Espacenet, and the PatFT patent databases.Inclusion and exclusion criteria: Inclusion: results obtained by searches using “machine learning ecg” or “machine learning ppg”. Exclusion: Articles published before 2018, articles that are not a review.Quality assessment (QA) checklist: Ensure that: i. The search strategy was correctly described; the inclusion and exclusion criteria were appropriate and well defined; the studies selected for inclusion were relevant to the research question posed. ii. The included papers were of high quality; the included studies were well designed and conducted. iii. The data were appropriate and well described; the methods used to include the papers were appropriate and well documented; the consistency of the results was adequately evaluated and presented.Data extraction form: i. By techniques used: ECG or PPG. ii. By type of ML algorithm used: linear regression, logistic regression, decision trees, random forest, or support vector machines.B.Conducting: Studies were gathered using the Mendeley reference management tool as a database.

Study selection and refinement: The searched literature items were checked for duplicates and filtered through the inclusion and exclusion criteria, leaving a set of articles selected for the paper.

## 3. Results

The search identified a total of 1686 articles and patents. The sources were divided into two groups: ECG and PGG. The full analysis according to the PRISMA 2020 methodology [16] is presented in the Appendix A as Appendix A. The paper presents the most important developments in the use of machine learning methods as a support for accurate ECG- and PPG-based heart rate analysis. It also outlines the most important research directions that should be pursued in the near future, including ensuring the accuracy of measurement and the real-time interpretation of the results, as well as validating the data in clinical trials.

## 4. Discussion

### 4.1. Sensors Empowered by AI—Basic Information

Medical sensors are devices that can provide information about vital, easily measured signals based on the measurement of biomedical parameters. Because of the complexity of these measurements and the range of possible interference, such measurements are complicated and require the accurate selection of signals from the sensor. This role can be fulfilled by machine learning. AI can also be used to capture data from medical sensors, based on which, disease changes can be predicted (Figure 1). For example, heart rate data obtained from wearable devices or from ECGs can provide information about heart health and may help predict certain types of disease. Examples include arrhythmias, which are abnormal heart rhythms that can be detected by analyzing HR variability and irregularity; or cardiac ischemia, in which changes in heart rate can signal decreased blood flow to the heart. Heart failure, which is characterized by the reduced ability of the heart to pump blood effectively, can be reflected in changes in HR patterns. Sleep apnea is a sleep disorder that can cause disturbances in breathing and HR patterns during sleep. Hypertension can also be detected, manifesting as elevated HR and changes in HR variability. The development of standardized and validated algorithms for analyzing HR data and accurately detecting disease markers remains a major methodological challenge. Of particular importance is the quality of the data; the accuracy of HR data is affected by various factors, such as the type of HR sensor, the position of the sensor, and the presence of motion artifacts. Ensuring high-quality HR data is therefore a critical challenge. Additionally, interpreting HR data in a clinical context requires a thorough understanding of the physiology and pathophysiology of heart disease, as well as a solid understanding of the limitations and strengths of HR sensors. Data privacy and security must also be ensured, as sensors contain sensitive personal information.

In terms of the sensors themselves, and in particular, a reduction in size, currently, most medical sensors are external and not wearable. However, there is a significant trend toward wearable sensors [17], making it possible to constantly control measured parameters without much interference in human functioning. It is interesting to note that such sensors are now being produced not only as watch-like meters, but also as contact lenses, mouth guards, and plasters [18,19,20]. The fact that such wearable sensors are becoming an interesting alternative to classical sensors is evidenced by the term “wearable sensing electronic systems” (WSES) appearing in the literature [21,22,23].

The research on medical sensors mainly focuses on the accuracy of measurement data readouts, the ability of these data to be quickly transformed into measurable values, and the rapid interpretation of the data [24]. Another important issue is the possibility of continuous measurement in real time as well as data storage [25,26,27]. Numerous studies have shown that medical sensors can exhibit a number of faults that cause erroneous readings [23,28]. AI support is proposed to solve these problems. It should be noted that when ANNs are applied to sensors, this should rather be considered machine learning (ML), which is a broader issue of artificial intelligence. AI and ML are closely related because ML is a subset of AI, as presented in Figure 2. ML is a type of artificial intelligence that allows systems to learn and improve from experience without the need to be explicitly programmed, as is the case with procedural computer languages. In this sense, AI is a broader term that encompasses technologies such as computer vision, natural language processing, and robotics, all of which use ML to achieve their goals. Moreover, deep learning (DL) is based on the structure and function of the human brain, and is specifically designed to handle large and complex datasets. DL is particularly well suited to tasks such as image and speech recognition, natural language processing, and autonomous decision-making. Generally, the process of how machine learning works can be described using the proposed scheme. Inputs refer to the first step in the machine learning process; that is, collecting and preparing the data that will be used to train the model. These data are often referred to as input or feature data. Algorithms are machine learning algorithms that are selected and applied to the input data. These algorithms use mathematical models to analyze the data and identify patterns and relationships. Finally, results refer to results obtained by applying the algorithm. An important outcome is that predictions or classifications must be generated from new, unseen data. The accuracy of the machine learning model depends on the quality and quantity of the input data, the choice of algorithm, and the parameters used to train the model.

Since readings from medical sensors must be accurate and inconsistencies must be excluded, various mechanisms have been developed to detect such errors, including those based on machine learning (ML). Thus, there is a potential opportunity to develop techniques using artificial neural networks to detect sensor faults. Faulty measurements not only decrease sensor accuracy, but can also lead to misdiagnosis, which can be life-threatening for the patient. Therefore, the ability to quickly detect erroneous readings and distinguish between minor inaccuracies of sensor damage is critical in increasing the reliability of measurements. Naive Bayes, Bayesian networks, Support Vector Machines (SVM), and Random Forests are commonly used algorithms for classification tasks, and Additive Regression techniques can be useful for prediction tasks in anomaly detection. Convolutional Neural Networks (CNNs) and recurrent neural networks are also effective techniques for anomaly detection, particularly in the context of image and sequence data [28,29,30]. All these techniques are considered part of AI because they are used to build models that can learn from data, make predictions, and improve over time. These techniques can be seen as part of the larger goal of AI, which is to build systems that can learn from data and make intelligent decisions or predictions. Naive Bayes (e.g., classification problems) and Bayesian Networks (e.g., reasoning under uncertainty) are part of the subfield of AI known as probabilistic reasoning or structured probabilistic models. SVM (finding optimal boundaries between different classes, typically separated by the hyperplanes in data space) and Random Forests (for classification and regression problems) are part of the subfield of AI known as supervised learning or machine learning. Convolutional neural networks (CNNs) are part of the subfield of AI known as deep learning, which is a type of machine learning that uses neural networks with multiple layers to learn representations of data. AI can also assist with data storage by saving disk storage capacity. For example, a hybrid prediction model based on density spatial clustering of applications with noise has been proposed. The aim is to remove outliers from sensor data and improve accuracy in detecting diabetic disease using the Random Forest machine learning classification technique [25].

Another advantage of AI is that this technology can be applied to a trend rapidly gaining popularity in medicine, namely, personalized medicine. This holistic approach to the patient allows for better treatment outcomes, and is supported by the use of ML techniques. For example Ba’s group [17] developed a system for extracting data from medical sensors with the personalized characteristics of older patients. This system is known as Machine Learning-Assisted Integrated Data-Driven Framework (MLA-IDDF). This customization increases compression detection performance with a reduced number of measurements. The procedure consists of the following steps: compression detection sampling, ML platform and training unit (using a CNN-based reconstruction network model), and signal recovery.

### 4.2. Types of HR Sensors Currently in Use (ECG [Including Holter] and PPG)

Noninvasive electrocardiogram examination is an essential tool for the evaluation and screening of cardiovascular disease (CVD). This clinical examination is important because statistics from the World Health Organization show that CVD is the leading cause of death worldwide, accounting for approximately 30% of all global deaths each year. By 2035, it is estimated that more than 130 million adults will have CVD symptoms [31]. Ambulatory electrocardiogram technology (known as the Holter electrocardiograph) is a pioneering noninvasive examination in the clinical cardiovascular field that monitors cardiac electrophysiological activity. Approximately 300 million ECGs are conducted [32] annually. It is well known that the ambulatory ECGs can detect non-sustained arrhythmias that are not easily detected by a routine ECG examination. In particular, ambulatory ECGs can increase the detection rate of transient arrhythmia and transient myocardial ischemic attack. The ECG graph consists of a sequence of waves (P, R, T), two intervals (PR, QT), and two typical segments (PR, ST) [33,34]. These parameters are essential for the clinical analysis of the disease, establishing the analysis, and evaluating the therapeutic effect. Ambulatory ECGs can determine whether symptoms, such as palpitation, dizziness, and fainting, are related to arrhythmia. Common types of arrhythmia detected by this technique include sinus bradycardia, conduction block, rapid atrial fibrillation, paroxysmal supraventricular tachycardia, and sustained ventricular tachycardia. Currently, ambulatory ECGs are considered one of the most important and widely used techniques for detecting these arrhythmias over a 24-h period. The old-fashioned digital signal analysis of ambulatory ECGs, which has become a classical analysis method in machine recognition methods for ambulatory ECGs, is based on the principles and rules described in 1985 by Pan and Tompkins [35]. The test results that refer to an arrhythmia standard database from the Massachusetts Institute of Technology show that the recognition accuracy rate of this method is 99.3% [36]. Many researchers have tried to employ other signal analysis methods, such as the wavelet transform method published by Cuiwei Li et al. [37], which is relatively influential. In summary, the following key drawbacks are related to the abovementioned conventional methods: (i) The anti-interference ability is relatively poor due to the effect of interference signals. The classification and recognition of the heartbeat cannot actively prevent the influence of interference fragments; (ii) It is impossible to determine the accurate extraction of features for the heartbeat. There is often excessive detection or missed detection of the heartbeat; (iii) The classification of the heartbeat stays in two types of sinus––supraventricular and ventricular, which is far from meeting the complicated and comprehensive analysis requirements of ECG doctors; (iv) Atrial flutter and atrial fibrillation, pacemaker heartbeat, and ST-T changes cannot be accurately recognized. Therefore, this is unable to assist patients with atrial flutter or atrial fibrillation, its usefulness in evaluating the functioning of pacemakers is limited, and it cannot accurately analyze the role of ST-T changes in myocardial ischemia; (v) The identification of the heartbeat and ECG events is not precise or comprehensive; therefore, heartbeats and ECG events will undoubtedly be missed due to the influence of many pre-existing factors, which will impact the interpretation of doctors; and (vi) The analysis methods do not objectively evaluate the signal quality of event segments, nor do they perform comprehensive analysis and statistics on the 24-h data. Report summaries and event screenshots still rely on the experience and ability of doctors. This can easily result in error, with the data not reflecting overall detection, missing report events, and poor or atypical screenshots of patient reports.

Thus, physicians must still spend a lot of precious time carefully reading the ambulatory ECG data. This means that the currently available equipment cannot effectively support their analysis ability, both in terms of quality and efficiency. Although most of the ambulatory ECG analysis software on the market can repeatedly analyze the data, due to the complexity and variability of the ambulatory ECG signals themselves, that data can easily be influenced by various interferences during the wearing process. Furthermore, for an average of about 100,000 heartbeats in a 24-h period, the accuracy rate of the current automatic analysis software is largely inadequate in helping physicians correctly analyze ECG data and provide accurate analysis reports in a relatively short time [38]. Therefore, it is proposed to use advanced scientific and technological methods, including AI technology, to help hospitals effectively improve the automation of ambulatory ECGs. Promising technologies have been proposed via different patents and/or patent applications. The patent entitled “Method and device for self-learning dynamic electrocardiography analysis employing artificial intelligence” demonstrates the feasibility of a device for self-learning, dynamic electrocardiography analysis using AI [39]. According to the patent declaration, the device allows for self-learning; fast and comprehensive analysis of the dynamic electrocardiography process; the recording of the electrocardiography process; and simultaneous data collection and analysis using a deep learning (DL) model. This solution ensures continuous training of the network, consequently increasing the accuracy of the analysis. The same concept of automatic electrocardiography analysis with the possibility of recording results, and modifying the data using DL is protected by the United States Patent Application 20200305799: “Artificial intelligence self-learning-based automatic electrocardiography analysis method and apparatus” [40]. Similarly, Kun and his co-inventors lodged the United States Patent Application 20200229771 for an explainable AI framework designed for ECG signal data analysis. This framework can be widely applied in ECG classification, computer-aided diagnosis, bedside alarms, and patient ECG monitoring [41]. This examination of patents aimed to provide a comprehensive account of ECG diagnosis and analysis systems to help doctors diagnose various heart diseases, including atrial fibrillation, myocardial infarction, and acute coronary syndrome (ACS). It is worth mentioning here that mathematical methods play an important role in solving technical problems in all fields of technology. However, these methods cannot be patented under Art. 52(2)(a). AI and ML are based on computational models and algorithms. Such computational models and algorithms are per se of an abstract mathematical nature, irrespective of whether they can be “trained” based on training data. When assessing the contribution made by a mathematical method to the technical aspects of an invention, it is necessary to consider to what extent the method has a technical purpose in the context of the invention. In such cases, AI can be classified as having the technical purpose of delivering a medical diagnosis by an automated system that processes physiological measurements. Therefore, most of the information about AI-based sensors can be found in the literature.

The cardiac rhythm captured by mobile or wearable equipment is typically assessed by one of two methods: electrocardiography (ECG) or photoplethysmography (PPG) [42,43,44]. Generally, the PPG technique is applied to mobile or wearable equipment, although units with ECG sensing are promising as well. PPG signals have the potential to be detected and measured from various parts of the body (e.g., finger, ear, wrist, arm) [45]. The PPG method, although based on different principles, is an excellent continuation of the ECG method, particularly in terms of reducing the size of the device and greater ease of controlling the parameters. A PPG signal is measured through optical sensors consisting of a light emitter [46] and a light receiver (photodetector) [47]. Such sensors can be configured to operate in either transmission or reflection mode. During pulse monitoring, light reflected from the surface of the skin is evaluated; the brightness of this light varies with fluctuations in blood pressure. However, an accurate HR determination is challenging if the PPG technique is used. To determine the heart rate, this method measures fluctuations in the reflected light intensity [48]. The effect of cardiac activity on light change is minimal––about 2%. A detector located on the forefinger generates a significantly more powerful output, but it is impractical to carry such a device. Therefore, consumers are more comfortable using wrist-mounted monitoring equipment, despite the output being considerably lower. Moreover, the development of this type of device is more demanding. There are numerous errors associated with this measurement technique because there are multiple underlying physical factors that interfere with correct readings. Such interference is primarily generated by motion of the user’s body (and thus, variations in blood circulation in the vascular system) and by movement of the sensor over the surface of the dermal tissue. However, very slight hand movements produce significant variations in light reflection when tested on the wrist. Another cause of error is the adhesion of the sensor to the skin at the strap, so that ambient external stray light also reaches the sensor. To compensate for the skewed sensor signal, various mathematical approaches can be used. However, it is extremely challenging to accurately isolate a signal related only to the HR. For example, an interference signal from a constant gait rate of one hundred steps per minute would be practically impossible to distinguish from a cardiac rate of one hundred bpm.

The key challenge of extracting features from noisy sensory data can be solved using ML methods. One promising method is to use convolutional kernel convolutional neural networks (CNNs) [49]. The kernel represents a weight matrix that is multiplied by the input data to extract the relevant features. The kernel travels through the data in the time domain. Each row shows the values of a time series for a certain axis. The kernel is only allowed to travel in one direction, lengthwise along the time axis, as shown in Figure 3. The CNN acts as an extracting agent of the data to be processed. The relevant characteristics can thus be extracted [50]. DL data-based pre-training is appropriate for long-term vital parameter measurements in everyday life. These are not influenced by local variations in vital parameters induced by exercise or by the intermittent starting and stopping of measurements when the measuring tool is recharged. Supplementary detectors are provided to reduce the influence of interference. For instance, accelerometers are used for motion recognition; thus, the reading is paused whenever intense body motion is sensed. However, specifically during vigorous physical activity, the user is mainly interested in motion measurement by measuring the pulse rate. Because of the numerous interferences and difficulties, the most commonly used measurement result is a multi-minute progressive time average.

### 4.3. ECG Sensors Empowered by ML

An analysis of the literature indicates that artificial neural networks can be used to analyze and classify electrocardiograms (ECGs) for the diagnosis and monitoring of heart disease. For this purpose, machine learning techniques, including deep learning models such as convolutional neural networks (CNNs) and long-short memory networks (LSTM), can be used. Among the most important trends in the improvement of such sensors is the development of novel methods or approaches to increase the accuracy of ECG classification. This has been achieved through the use of new models, optimization algorithms, and semi-supervised learning methods.

An innovative approach to analyze ECG signals for the diagnosis of heart disease is through the algorithm proposed by Aziz et al. [34]. The algorithm combines two techniques—two event-related moving averages (TERMA) and fractional-Fourier-transform (FrFT)—to detect P, QRS, and T waveform peaks in ECG signals. The TERMA algorithm is used to locate the desired peaks, while the FrFT is used to identify the locations of various peaks in the time-frequency plane. After peak detection, the authors used the estimated peaks, the duration between different peaks, and other ECG signal features to train an ML model for the automatic classification of heart diseases. Two different machine learning algorithms—the Multi-Layer Perceptron (MLP) and the Support Vector Machine (SVM)—were used to classify the ECG signals into different categories of heart diseases. The ML model was trained on the Shaoxing People’s Hospital database, which contains more than 10,000 records, and was tested on two other databases, with promising results. The authors demonstrated that the MLP classifier performs better than the SVM classifier for a variety of ECG signals. They also performed the challenging task of cross-database training and testing, and reported promising results. The overall accuracy of the trained model on two different databases was 99.85% and 68%, respectively. Alfaras et al. [51] proposed a fully automatic, rapid ECG arrhythmia classifier based on Echo State Networks (ESN). This is a simple brain-inspired machine learning approach that uses raw ECG waveforms and time intervals between heartbeats as input. The classifier was trained and validated through an inter-patient procedure, and required only a single ECG lead for feature processing. The classifier was evaluated using two ECG databases—the MIT-BIH AR and the AHA. It achieved comparable results with state-of-the-art, fully automatic ECG classifiers, even outperforming some classifiers with more complex feature selection approaches. The heartbeat classifier is designed to classify heartbeats into two classes based on morphology: SVEB+ and VEB+. The SVEB+ class includes normal (N) and supraventricular ectopic (S or SVEB) heartbeats, while the VEB+ class comprises ventricular ectopic beats (V or VEB) and fusion beats (F). The classifier is built on an ESN with a ring topology, which is a popular application of Reservoir Computing (RC)—a computing paradigm that has been successfully applied in various tasks. The ESN consists of three layers—input, reservoir, and output. The reservoir is a recurrent neural network with random input and connection weights. The ESN has two main advantages over other classical methods, such as the SVM, NN, or decision trees. This is because past heartbeats play a role in the classification task due to its intrinsic memory, thereby improving performance.

An example of the use of a large database is provided by Śmigiel [33], who discusses the use of ECG signal monitoring systems to evaluate and diagnose heart disease. The study was carried out on the PTB-XL database, which contains 21,837 records of 10-s ECG recordings annotated by cardiologists. The database has 71 types of heart disease and 5 classes, including normal ECG, myocardial infarction, ST/T change, conduction abnormalities, and hypertrophy. The authors used orthogonal matching pursuit algorithms and classical ML classifiers to develop cardiovascular disease classification models, with a new method to detect R waves and determine the location of QRS complexes. They also developed novel methods of ECG signal fragments containing QRS segments. The implementation of classification issues achieved the highest accuracy of 90.4% to recognize 2 classes, less than 78% for 5 classes, and 71% for 15 classes. The study aimed to find the optimal classification model for different classes using Feature Selection Methods. The methodology involved filtering the records; labeling and segmenting R peaks; dividing the data into training, validation, and test data; creating dictionaries; performing an Orthogonal Matching Pursuit operation; and evaluating the effectiveness of the proposed network methods. The authors suggest that ECG signals subjected to R peak detection, QRS complexes extraction, and resampling perform well in classification using decision trees due to signal structuring.

Wang et al. [52] proposed a new wireless ECG patch for remote cardiac monitoring. They also proposed a deep learning framework based on the CNN and LSTM models for ECG classification as normal heartbeat, ventricular premature beat, supraventricular premature beat, atrial fibrillation, and unclassifiable beat. The approach uses a semi-supervised method with confidence level-based training to process badly labelled data samples and improve accuracy. The proposed model for ECG classification is based on a residual network (ResNet) with newly designed confidence level-based training. The typical types of ECG signals were considered: P and T waves; QRS complex; and PR, QT, and ST intervals. The model consisted of six residual blocks, with the number of filters doubling for each block. The input to the model was a zero-mean unit variance raw ECG signal segment truncated to the first minute. This was because CNNs require a fixed window size. The residual block is the building block of the network backbone; it consists of two convolutional layers and an identity connection from the input to the output. The identity connection passes more information from the shallower layer to the deeper layer by a simple addition operation, enabling deeper learning networks. The confidence level is defined as the output probability of the last layer of the model, and a threshold is set to consider only clean data for training. This approach is called confidence level-based training, and it helps stop the model from being impaired. The proposed approach achieved an average accuracy of 90.2%, which is 5.4% higher than conventional ECG classification methods.

The following two studies also focus on the ML techniques, but focus specifically on popular and relatively cheap electronic devices such as Arduino. Shrestha et al. [53] aimed to develop a mobile health system that enables users to monitor and analyze their heart rhythm in real time using IoT techniques and ML. The proposed system used an AD8232 sensor to collect ECG signals (P and T waves, QRS complex). These were then sent to a minicomputer (Raspberry Pi) through an ADC converter, while a Flask application sent the digital ECG signals to a trained ML model for classification. The algorithm used the Random Forest machine learning model to classify the heartbeat. The researchers chose this model because their previous work found that Random Forest performed the best in predicting ECG data among seven other ML methods. The model was trained on three different heart rhythms from the PhysioNet, including normal sinus rhythm, congestive heart failure, and atrial fibrillation. The training and testing processes were performed on the IoT device itself. The dataset was divided into equal segments for each heartbeat type, and 500 segments were created for each type. Each segment contained 20,000 data points, and a label was attached to indicate the type of heartbeat. The dataset was randomized and split into training (70%) and testing (30%) subsets. The trained Random Forest model was then used to predict the heartbeat type of the real-time ECG signals collected from the Raspberry Pi using the ECG sensor. The authors state that the use of ML allowed the system to classify heartbeat rhythms accurately and in real time, allowing users to monitor their heart rhythm at home. Singh et al. [54] proposed an ECG monitoring (P, QRS, and T waveform peaks) and analysis system to remotely monitor patients with heart disease. The system used machine learning models, such as k-Nearest Neighbors, Random Forest, Decision Trees, and Support Vector Machines, to detect and analyse heart disease. The analysis was divided into several phases, namely, dataset retrieval, preprocessing, feature selection, feature reduction, and implementing ML models with different performance measures to evaluate the precision of the different models used. The dataset used in the study was taken from the MIT-BIH dataset. The data were divided into two vectors: one with data without the arrhythmia column and the other with data from the arrhythmia column. These were then fed into selected classification models for training. The test data consisted of around 700 records with parameters such as amplitude, RR, speed, age, sex, and medication. System performance was evaluated using various measures, such as precision, recall, F1-score, and accuracy. The proposed system was a standalone smart ECG system integrated with a Django web application. The application was connected to an IoT module to receive ECG signals in real time using Firebase. The ECG readings were monitored and analyzed using the machine learning model, and the results were represented in the Django web application. The system was designed to be low cost, making it an effective solution for the remote monitoring of patients in low-infrastructure areas. The system results indicate that the proposed system integrates real-time monitoring and analysis of the ECG signals at a low cost with an accuracy of 99%, making it more effective than existing systems. Future development of the system includes adding more data with different arrhythmia types, upgrading the IoT module to reduce noise in single-lead sensors, and sending prior notification to the doctor and emergency contact list in case of a detected medical emergency.

The number of leads and time of records are important considerations when using heart rate sensors. This is because these factors can impact the accuracy and detail of the information obtained, and can help clinicians make more informed decisions about a patient’s heart health. The more leads used, the more detailed the information obtained about the heart’s function. Thus, a 12-lead electrocardiogram (ECG) uses 10 electrodes to record the heart’s electrical activity from 12 different angles. This provides a more comprehensive view of the heart’s activity than a 3-lead ECG, which is most commonly used to record a 24-h reading. In general, authors use available databases in which the number of leads for a given dataset is specified depending on the actual sample, and this varies between records. However, such detailed information is rare for training artificial learning systems. Therefore, it is necessary to consider this aspect in future research on cardiac sensors supported by machine learning.

### 4.4. PPG Sensors Empowered by ML

Determining HR based on PPG data can be conducted using two basic methods. The first method involves analyzing the time domain PPG waveform through peak detection and period computation of the pulse. The second method uses frequency domain analysis, in which the dominant frequencies of the underlying PPG waveform are monitored. Subsequently, the potentially clearest frequency becomes the outcome of the heartbeat estimate. However, frequency analysis is also capable of considering non-linear time series. The wavelet transformation for the heart rate and breathing rate from the PPG signal or signal representation is the time-frequency spectrum that can be used [55]. For example, the literature suggests a model-based, two-stage algorithm consisting of movement-related noise artifact elimination and spectral-based analysis [56]. Motion-related artifacts are removed using acceleration information, while spectral analysis of the signal allows the extraction of spectral peaks matching the cardiac rate. The spectral power density of the PPG (waveform with alternating current [AC] and direct current [DC] components) and the accelerometric signals are calculated in another algorithm. This algorithm is known as the Spectral Filter Algorithm for Motion Artifact and Pulse Reconstruction (SpaMA) [57]. The AC component, also known as the pulsatile component, corresponds to variations in blood volume in synchronization with the heartbeat. The DC component results from reflected or transmitted optical signals, and is determined by venous and arterial blood as well as the tissue structure. The DC component presents minor changes with respiration, while the basic frequency of the AC component changes with the heart rate, and is superimposed on the baseline DC. By comparing the PPG and accelerometer spectra, spurious maxima in the PPG signal spectrum can be removed on the basis of the maxima in the accelerometer spectrum. Some older approaches [58,59] have been reported, in which the developers isolated the respiratory contribution from the PPG waveform by applying Fourier analysis.

A time-domain technique has been proposed for adjusting the measured PPG signal (light transmitter and receiver [photodetector] signals) [60] by decreasing the interference caused by body motion. Time series prediction ML using a trained LSTM was used to rectify the actual originally reflected PPG light from the skin, supported by auxiliary signals acquired from a triaxial accelerometer. The application of a suitably trained neural network (NN) to correct the PPG signal is a novel method. This method represents an innovation on traditional methods, for example in [61,62], where adaptive filtering was used. LSTM NN is a method with high application potential for time-series forecasting. This method is expected to be effective in reducing motion-induced noise distortion of the PPG signal. Using this method, it is possible to use actual and prior data obtained from the PPG signal and acceleration as input. The triaxial accelerometer sensor data are applied to obtain data about the individual’s movement, and the PPG signal from the light detector is adjusted based on this information. The adjustment is performed by a suitably trained LSTM. For the PPG signal enhancement reported in this paper, the LSTM network was deployed using TensorFlow software [63] using the Keras libraries [64] developed in the Python programming language [65]. Technically, the network was assembled programmatically using the Sequential class of Keras. This arranges a stack of network layers into a model that provides training and inference functions. ML inference refers to the workflow of inputting given data items into an ML model to compute a data output, for example, a numerical result. The resultant model is composed of an input layer, with one or possibly two LSTM layers and a terminating Dense layer. The addition of a greater number of layers did not yield an advantage; therefore, in future studies, only one- and two-layer networks should be used.

The networks that were trained were tested on the most extensive collection of data available to the authors, namely, the PPG-DaLia database [55]. This database includes over 35 h of stored data recorded from 15 participants. Included in the database are PPG (time-frequency spectra) and accelerometry signals, with the corresponding ECG data taken to serve as ground truth. The processed data signals in this database were gathered while performing eight distinct common everyday tasks under supervised conditions, similar to real life. Process signals from the PPG-DaLia dataset (PPG and acceleration) were submitted to trained LSTM networks. The outcome of the trained networks comprising the adjusted PPG waveforms was parsed through the peak recognition algorithm taken from previous work by the authors [66]. Of the LSTM network variants studied, several of the setups showed decent efficiency. It should be noted that comparable performance was observed for variants that required both low and high computing workloads. Thus, the research reported in the study should assist in determining appropriate scenarios for a given implementation and accessible computational means. It is evident from the calculations that the LSTM network substantially boosts the performance of the Time-Domain Heart Rate (TDHR) algorithm, thereby improving its efficiency compared with other algorithm variants analyzed. A widely debated problem is the relatively high complexity of algorithms based on the use of NN, which can translate into high energy consumption of mobile devices. At this point, hardware solutions based on the hardware implementation of a pre-trained network can be suggested. This opens up an area for application, but this is a separate problem that remains to be solved.

Another important problem related to the use of NNs in general, and in the case of biosensors in particular, is the availability of training data. One of the main challenges is the small amount of clinically validated outcome data for ML models. The scarcity of high-quality data that can be used for evaluation translates into the quality of the performance of these models in real-world clinical settings. This is particularly important because decisions made by ML algorithms can have an impact on patient outcomes, and having solid evidence to support their use is critical. Another challenge is the issue of data representativeness. ML models are only as good as the data on which they were trained, and if the training data are not representative of the selected patient population, the quality of the model can be seriously compromised. For example, if a model was trained on data from a specific demographic group or clinical setting, it may not perform well when applied to patients from other groups or settings. In addition, the issue of representativeness can also lead to biased models in which certain groups of patients may be systematically underrepresented or overrepresented in the training data. This can lead to unfair or inaccurate decisions made by the model, which can have serious consequences for patient care. The PPG-DaLia database mentioned above is an answer to this problem, as well as other sources [57,67,68,69]. An example is the PPG-DaLia dataset for motion compensation and heart rate estimation in activities of daily living. This dataset includes more than 36 h of data in total, recorded from 15 subjects. The data collection protocol included eight different types of activities (which are typically performed in daily life) as well as the transition periods between the activities. The subjects performed these activities under natural, close to real life conditions. The public can access the PPG-DaLia dataset and use it to estimate heart rates based on photoplethysmography. The dataset includes data from both wrist- and chest-worn devices, and covers a variety of physical activities that are performed under conditions similar to real life. The data are multimodal, meaning they include physiological and motion data. The dataset also includes electrocardiogram (ECG) data, which provide accurate heart rate measurements. Using the PPG and 3D-accelerometer data, users can estimate heart rates while compensating for motion-related distortions. The proper design of any AI system requires access to training data, which helps to address the problem of training and validating a given network. DL networks are a special case since their inherent design requires huge datasets. However, in most cases, biosensors do not require such a complex approach as the one required for DL networks. This is because most of the necessary data are categorized by the designer, who creates the solution. In most cases, there is no need for biosensors to be equipped with self-determining elements on how to categorize the data.

The previously mentioned motion artifacts (MAs) are severe contributors of noise to PPG data, considerably influencing the assessment of the HR and various other physiological variables. Another method known as SPECTRAP [70] can be implemented for precise movement-proof HR estimation, achieved by using PPG and parallel acceleration signals. The method uses a novel spectrum subtraction technique consisting of the following steps: i. calculating the PPG and acceleration signal spectrum, and ii. Subtracting the MA component of the spectrum from the PPG signal. This spectrum subtracting approach relies on the asymmetrical least squares method, avoiding the shortcomings of classical spectrum subtraction techniques. In order to locate the spectral peak matching the HR on the outcome spectrum, SPECTRAP addresses the issue as a pattern classifier type problem and employs Bayes decision theory to resolve the issue. From the experimental results provided on the PPG database employed in the 2015 IEEE Signal Processing Cup, it was demonstrated that the suggested algorithm possesses superior efficiency. The mean absolute error across twelve training datasets was 1.50 BPM (SD: 1.95 BPM), while among the ten testing sets, it was 2.13 BPM (SD: 2.77 BPM).

It is clear that PPG signals are used extensively for HR tracking. Indeed, relative to the electrocardiogram, PPG signals in general can be conveniently obtained using wearable equipment, such as smart watches, and at low cost. PPG data signals are frequently affected by movement artifacts (MAs) and inferred noise, both of which significantly decrease signal fidelity and create serious issues for HR tracking. Another algorithm based on spectral subtraction and NN has been proposed for precise HR monitoring where MA and noise are present [71]. In particular, the MA component of the spectrum is evaluated using acceleration signals (ACC), and is subsequently extracted from the PPG spectrum. In a further step, an NN is modelled on the basis of the new characteristics extracted from the ACC signals. This is used to detect the correlation between changes in ACC and HR in subsequent time frames. This feedback is then applied as a comparison to choose the spectral peak that matches the real HR. A data post-processing approach is adopted to rectify incorrectly identified HR and enhance precision. Based on the NN, the algorithm is verified against the 2015 IEEE Signal Processing Cup Dataset, reaching a mean absolute error of 1.03 BPM (SD: 1.82 BPM). The authors conclude that it surpasses earlier efforts found in the data literature.

There are also reports on the application of DL algorithms in cardiovascular monitoring devices. The “PP-Net” DL algorithm presented in [72] is able to obtain estimates of the following physiological characteristics: Diastolic Blood Pressure (DBP), Systolic Blood Pressure (SBP), and HR using a single network and with a single-channel PPG signal. This model was developed using a DL Long-term Recurrent Convolutional Network (LRCN), demonstrating an intrinsic characteristic extraction capability. This allows it to avoid expensive feature screening and extraction steps, which, in turn, decreases implementation complexity in limited resource computing systems, such as mobile device platforms. The PP-Net’s efficiency was demonstrated on the broader and openly accessible MIMIC-II database. The Multi-parameter Intelligent Monitoring in Intensive Care (MIMIC-II) database is the largest database of its kind. The dataset consists of simultaneous recordings of multiple physiological signals and parameters from intensive care unit (ICU) patients, including electrocardiogram (ECG), photoplethysmograph (PPG), and arterial blood pressure (ABP). The model attained a normalized mean absolute error of 0.09 (DBP) and 0.04 (SBP) mmHg for BP, and 0.046 bpm value to estimate HR on the entire sample population of 1557 critically ill patients. The authors conclude that the accurate assessment of HR and BP on a wider sample population demonstrated the efficiency of the suggested DL approach framework compared with previous methods. Accurate estimation in a massive population presenting cardiovascular complications further validates the reliability of the suggested framework in health care pervasive tracking, particularly in monitoring cardiac and stroke rehabilitation.

Technology giants such as Google and Microsoft offer tools to support the development of NN technology. These platforms are also used in biosensors supported by AI algorithms. Another accelerometer-based removal of MA from PPGs was investigated in the work of [73], following a number of studies previously based on DL algorithms. Different preprocessing approaches were benchmarked, optimal preprocessing parameters were determined, and efficiency was improved by applying a model tuning approach. Furthermore, the underlying model was specifically optimized using hyperparameter lookup and neural based architecture using Neural Network Intelligence provided by Microsoft. This model was composed of 3 staggered 1D convolutional layers with Batch normalization, 2 LSTM layers, 1 fully connected layer, and a SoftMax function. This is a multi-dimensional generalization of the logistic function. The size of the model was 6 × 256 × 4, achieved by combining six steps of 256 × 4 each, derived from PPG and ACC preprocessing. The outcome was a 75-fold decrease in parameters compared with previous work, and a 26% enhancement in the mean absolute error, decreasing from 7.65 to 6.02 BPM. The authors also used the PPG-DaLia database to benchmark the engineered solution.

### 4.5. Ongoing Challenges

The successful application of ML methods requires, first of all, a large amount of data. Without it, there is no way to properly train the ANN. This is not just a problem with the medical sensors described in this article, but a general problem that arises when using artificial neural networks. Therefore, the implementation of such technology requires large amounts of data or incurring the cost of purchasing it.

Key technological barriers still need to be overcome to enable and accelerate the commercialization of new biosensing device technologies. First, there is a need to significantly minimize the hardware so that it can operate with minimal power requirements and provide continuous diagnostics. Another important issue is the communication of sensor measurement data. Near Field Communication (NFC) is often proposed in this regard, as well as power supply. However, this technique has weaknesses, nor does it support continuous monitoring or data transfer, which are vital in medical sensors. Moreover, charging can be problematic given power limitations, which makes the charging slow. Among other things, self-powered devices based on triboelectric nanogenerators (TENGs) are becoming the answer to this problem [74,75,76,77]. These sensors have the ability to collect large amounts of data over a long period of time. Moreover, TENGs have shown promise as an energy harvesting technology through their high output performance and ability to efficiently transfer surrounding mechanical stimuli into electricity. Triboelectric nanogenerators can also actively function as self-powered sensors. Ongoing research indicates that the use of such devices in conjunction with AI could have applications in precision cancer research for cancer management and prevention [78]. This, and many other examples, indicate that the problem of power supply and data collection is likely to be solved in the near future.

At this point, it is also worth mentioning efforts to solve the problem of large amounts of data from sensors and, in particular, the transfer of these data to the computing unit. Intuitively, it was proposed that the best solution would be to compute as much information as possible within the sensor. This would ensure the efficient processing of large amounts of data and reduce power consumption. Such concepts are known as near-sensor and in-sensor computing, in which computational tasks are partially moved to the sensor terminals. Although such a solution has not yet been used in medical sensors, it is a potential solution for hardware implementation of integrated sensing and processing units. This solution uses advanced AI technologies, including convolutional and spiking NNs. However, it should be noted that although this computation has high potential, it is still in the very early stages of development [79].

Another challenge and opportunity arising from the collection and processing of data from medical sensors by AI is the ability to connect this solution to the Internet of Things (IoT) [17,80,81,82]. This solution allows the remote monitoring of a patient’s health, for example, from home. This smart solution of an IoT-based diagnosis system seems to be optimal because it allows for continuous contact with the patient, especially in cases of chronic disease. Such automatic monitoring of vital signs allows for the diagnosis of diseases, while AI support allows for rapid access to data and support of the diagnosis. Connected wearable devices and IoT can transform the medical system. However, the integration and design of wearable sensors poses many challenges, especially in the areas of data sharing, patient monitoring, and diagnosis, including the selection of appropriate ML techniques [25,83].

The social aspect is also important. No medical sensor with AI will be used if patients do not accept such a solution. Fear of novelty is always present and steps should be taken to verify the effectiveness and safety of such a solution. Research in this area has shown that it is often necessary to involve family and friends to convince people to use technological solutions, especially older people [84].

Although the role of AI cannot be overestimated and it is always at the end that the effectiveness of an operation can be confirmed, it is undoubtedly a breakthrough in relation to sensors based on AI.

## 5. Conclusions

An analysis of recent developments in the field of cardiac sensors indicates that the application of machine learning techniques is highly desirable. Such support has been shown to improve sensor accuracy. ML algorithms can analyze large amounts of medical sensor data to detect subtle changes in heart function. These may indicate the presence of cardiovascular conditions, which can lead to more accurate diagnoses and treatment decisions. Furthermore, ML algorithms can be used to analyze medical sensor data in real time, enabling health care providers to remotely monitor patients and quickly intervene in the event of a cardiac emergency. Therefore, it seems that ML in the field of sensors will continue evolving. In terms of social behavior, people want technological innovations and are becoming less and less afraid to use them. Therefore, trust in such solutions is growing. Although there are currently no widely available AI-based medical sensors on the market, the research presented in this article, as well as the successful application of AI in other fields, indicates that AI will probably become an invaluable tool in medical sensors in the near future. Looking ahead, the availability of ML-supported heart sensors is likely to support personalized care. The ability of ML techniques to learn and identify could support the analysis of medical heart sensor data. This can provide personalized health insights and recommendations based on a person’s unique cardiovascular profile, health history, and lifestyle factors. These sensors will undoubtedly contribute to better control and optimization of the treatment process, thereby saving human lives.

## Figures and Tables

**Figure 1 ijerph-20-04605-f001:**
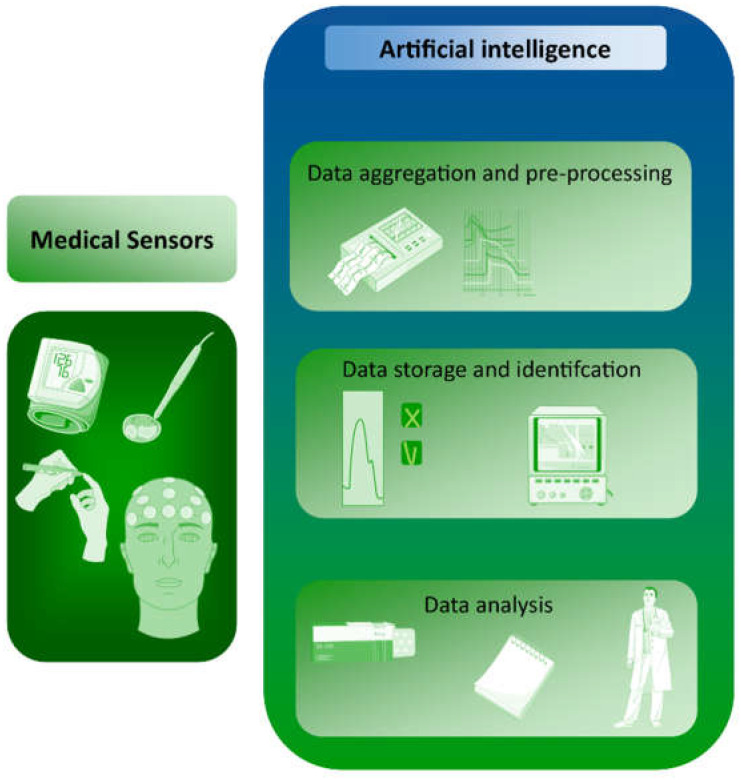
Applications of AI in medical sensors, from data collection to diagnosis (medical art from Smart Servier Medical Art. Available online https://smart.servier.com (accessed on 1 December 2022)).

**Figure 2 ijerph-20-04605-f002:**
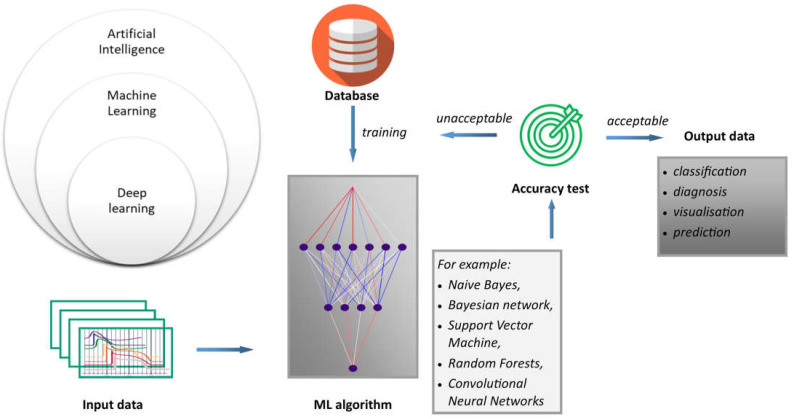
General scheme of data processing.

**Figure 3 ijerph-20-04605-f003:**
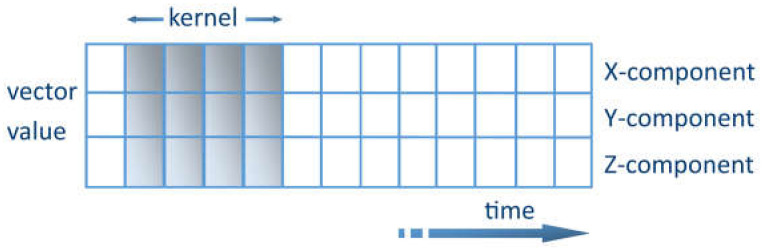
Representation of the kernel with 3D vector quantity.

## Data Availability

Not applicable.

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
