# Peer review of "From Data to Diagnosis: How Machine Learning Is Changing Heart Health Monitoring"

_ijerph, 2023, doi:10.3390/ijerph20054605_

Round 1

Reviewer 1 Report

Comments:

Thank you for the opportunity to review this work.

This is a very well researched paper that describes the current landscape regarding HR detection and tracking, the attempts to utilise ML to improve HR detection and tracking, and the current limitations.

The authors find that:

- there are currently no AI based medical sensors on the market

- AI will probably become an invaluable tool in medical sensors in the near future

- these sensors should undoubtedly contribute to better control and optimization of the treatment process and saving human lives.

I can see that a huge amount of work has gone into the preparation of this paper and the authors are to be congratulated for their efforts. However, I think there are significant issues with the conclusions of this paper, which must be addressed. Additionally, I think that this paper would benefit from both some structural changes and further clarifications, in order to more clearly define the scope of the paper and the findings.

Major issues:

1.     The introduction seems to imply that the paper will focus on HR detection and tracking and specifically investigate the potential for AI supported HR monitoring to replace the somewhat cumbersome Holter monitoring. However, at times the paper discusses cardiac tracing via ECG that is beyond the scope of HR tracking (e.g. sinus bradycardia, conduction block and ST changes). Please make the scope of the paper be clear in the introduction and consistent throughout the paper.

2.     Personally, I do not like the use of the terminology AI. To many readers, this gives the impression of a ‘thinking’ entity rather than a trained algorithm. My recommendation would be to use the terminology ‘machine learning’. However, if you choose to use the AI terminology, please include a clear definition of what AI is (and what it is not). This definition should be specific to the clinical context (and therefore explicitly call out the relationship between AI and ML).

I would recommend including a diagram that shows how ML works

i.e. inputs  algorithm  outcomes

This makes it clearer to the reader what is required for a model to work (i.e. data must be provided for inputs, e.g. ECG trace data, and outcomes, e.g. HR, and that the point of the algorithm is to correctly predict the outcome based on the input)

3.     The authors state that medical sensors are the key to properly assessing human health.

This statement requires expansion. In what way do medical sensors hold the key to properly assessing human health? To date, sensors have only been deployed in the case of sickness. What evidence to we have that monitoring of the general population who are deemed ‘healthy’ will be of benefit?

This is a crucial translation question that must be answered given the proposed applications of AI supported sensors.

4.     I think the structure of the paper would benefit from clearer headings. I would suggest the following under section 2:

a.      Types of HR sensors in use currently (ECG (including Holter), PPG)

b.     Limitations of HR sensors

                                               i.     signal integrity (measurement issue) (e.g. due to motion artefacts)

                                              ii.     signal interpretation (clinical issue)

c.      Potential applications of ML

                                               i.     signal noise reduction and improved accuracy

                                              ii.     diagnostic decision support (i.e. predicting diagnoses based on signals)

Once these heading are clear I would recommend explicitly stating whether both of the above potential applications are within the scope of the paper. The majority of the content appears to be related to signal integrity improvement. If the authors decide that diagnostic decision support is within scope, they need to include a section dedicated to this and be very clear about the limitations of using sensor data in this way (i.e. lack of ‘healthy’ training data, lack of clinical outcome data, need for model performance well beyond what is usually required for ML projects etc)

5.     Under section 2.1 the authors claim that AI can be used to collect data from medical sensors and on that basis predict disease changes and they provide a figure by way of example.

Prediction of disease based on sensor data is complex and requires access to high level clinical outcome data. If authors want to include this statement, please

a)     provide concrete examples of the types of diseases that can be predicted from sensor data, and  

b)     the methodological, data and clinical challenges that must be overcome to realise this.

6.     Authors state that: ‘Anomaly detection can be successfully supported by algorithms such as Naive Bayes, Bayesian network, Support Vector Machine (SVM), and Random Forests algorithm for classification jobs and Additive Regression techniques for prediction jobs for anomaly 80 detection, Convolutional Neural Networks (CNNs) and recurrent neural networks.’

Please state how these techniques fit in with ‘AI’ as defined by the authors? (i.e. when AI is defined, these specific methodologies should be named and defined too)

7.     Authors state that: ‘the ambulatory ECG can find whether symptoms including palpitation, dizziness, and fainting of patients are related to arrhythmia, such as sinus bradycardia, conduction block, rapid atrial fibrillation, paroxysmal supraventricular tachycardia or sustained ventricular tachycardia, which currently is considered one of the most important and most widely used situations of the 24-hour ambulatory ECG.’

This sentence requires rewording as it is not accurate. The ECG can only record arrhythmia. It is the clinician who interprets whether the condition is connected to the symptoms, and this requires a detailed timeline not just for the cardiac trace (which the ECG provides) but also the symptoms (e.g. palpitations, dizziness and fainting), which the patient must provide.

An ambulatory ECG can ‘catch’ arrhythmias but the determination of whether the arrythmia explains the symptoms is largely clinical. Many people have transient arrythmias that do not explain their symptoms and another clinical cause is found.

It is important that the function of the ambulatory ECG – i.e. to collect cardiac trace data – is not confused with the clinical application of the ambulatory ECG data.

It is important to distinguish where the data is coming from so as not to give the impression that the ambulatory ECG has all the pieces required to make a diagnoses.

8.     In the section about ambulatory ECG monitors, the current clinical context of their use has not been addressed. In general, these are provided to patients in the short term in order to ‘catch’ the offending pathology, e.g. AF, which is usually intermittent.

Use of ambulatory ECG monitors in the general public is well outside the current scope.

The authors seem to suggest that

a) ambulatory ECGs could be utilised by the general public in the future and that

b) this constant monitoring could be used to diagnose an acute event (e.g. a myocardial infarct).

Given that both of these potential applications are well outside the current usage of ambulatory ECG monitors, the authors should clearly state the current use, their idea of future use, and the limitations and challenges that must be overcome to realise their suggested future.

9.     Page 7 includes an important section summarising issues with training data. However, this is not the only data limitation. Clinically validated outcome data is also lacking, and there are issues regarding the representativeness of the training data that is available. These data limitations should also be discussed.

10.   The authors do not state whether the PPG-DaLia data was collected from the general public or from a hospital population. This should be rectified.

11.   With regard to the "PP-Net" study, please describe in brief the relevant methodology. Please ensure to include a description of what data was used.

e.g. MIMIC-II databank includes a large amount of data including time series and longitudinal data. Did the study use heart rate to predict the SBP and DBP at the time of the HR measured (or the BP within 15 minute of HR monitoring etc)

Other important questions include whether baseline BP was already known. Since HR and BP are part of the cardiac output algorithm, it  makes a significant difference to the interpretation of the model if baseline BP is known.

What other inputs were included (e.g. sex, age))?

Please also state clearly for the reader whether the MIMIC-II databank includes data from a hospital population or the general population and this is essential for interpretation.

12.   The authors state on page 9 that: ‘Finally, a clinical study conducted on piglets connected to blood glucose monitoring sensors with AI-based software demonstrated the superiority of such a solution over an experienced ICU physician.’

Inclusion of this reference requires further context. Blood glucose monitoring is a process with defined, simple input (BGL + insulin)) and a defined simple outcome (hypoglycaemia). The clinical context is vastly different to HR monitoring and the two should not be confused. There are very good reasons why ML supported glucose control is effective and many of these do not apply to ML supported physiological monitoring.

If authors want to include this reference, they must highlight how the clinical context of BGL monitoring relates to HR detection and properly contextualise the study.

13.   In general, I think it is important for the authors to consider and address the question of what is the purpose of measuring HR in asymptomatic people?

The authors provide a good case for the utility of ML in improving sensor data integrity by reducing noise and MA. However, the context of this use remains unclear. If the authors want to imply that monitoring of HR could be used in the general public, then they must answer the above question.

Minor issues:

1.     At line 103 there is a typo: The breakthrough ambulatory electrocardiogram technology (known as Holter electrocardiograph) is considerate

This should be ‘considered’

2.     At line 208 there is a typo: An further cause of

Reviewer 2 Report

The novelty of your contribution is missing. The 3 figures not own contributions.

Reviewer 3 Report

The article is devoted to the latest developments in heart rate sensors, supplemented by artificial neural network procedures. The importance of using artificial intelligence in the field of using medical sensors in medical diagnostics in the field of collecting and processing data and interpreting the results is indicated. The most critical problems and prospects in this area are shown.

The paper has some shortcomings that need to be corrected.

  1. The abstract should be expanded with the methodology and findings of the research.
  2. Sources 4-15 are unnecessary. It is unclear why the authors select those papers to illustrate the role of AI in medicine.
  3. The aim of the paper should be defined.
  4. The review methodology should be described.
  5. The inclusion and exclusion criteria should be defined.
  6. The text is unstructured. It is recommended to separate it by some features, e.g., tasks, approaches, etc.
  7. Authors describe only heart rate sensors, but many other sensors are used in healthcare.
  8. Regarding heart rate sensors, the authors do not discuss the number of leads and time of records.
  9. A discussion is needed.
  10. The novelty of the research and its contribution to the field should be highlighted.

In summarizing my comments, I think the paper does not fit the journal and can not be published in its present form. 

Reviewer 4 Report

Summary:

The authors provide a systematic review of the recent developments in heart rate sensors and artificial neural networks.

General concept comments: 

This article provides a review of recent works in heart rate sensors, artificial neural networks, and ongoing challenges. The article may be enhanced by providing the following:

  • Summary of related review works in heart rate sensors

  • Review questions and rationale for synthesizing the empirical evidence for this systematic research work

  • Methodology employed for reviewing the literature (for example, PRISMA methodology)

  • Search strategy (Google scholar, pubmed, etc)

  • Literature inclusion/exclusion criteria

  • Tabular summary of articles in section 2.2 

I recommend accepting the article after the comments are satisfactorily addressed.

Round 2

Reviewer 2 Report

The papers looks much better now. Could you please think about the titel here are some creative names:

  1. Advancing Healthcare with Machine Learning: The Future of Heart Sensor Technology
  2. From Data to Diagnosis: How Machine Learning is Changing Heart Health Monitoring

Please improve the argumentation and the grammar in the introduction and conclusion section.

Reviewer 3 Report

Thanks to the authors for the corrections made to the article. However, the article is still unsuitable for publication in this journal.

1. The search methodology does not allow repeating the study. A search string, a straightforward algorithm for including articles in the study, and inclusion and exclusion criteria should be included. The phrase "with a few exceptions of articles essential to the development of the techniques described" is unclear who and by what criteria determined the article's importance.

2. The authors focus on cardiac sensors while not considering in detail what problems their implementation solves.

3. The authors focus on machine learning and artificial intelligence while not paying enough attention to the data that can be collected using sensor data. However, data is one of the essential aspects.
